# Impact of *Escherichia coli* and Lipopolysaccharide on the MAPK Signaling Pathway, MMPs, TIMPs, and the uPA System in Bovine Mammary Epithelial Cells

**DOI:** 10.3390/ijms26083893

**Published:** 2025-04-20

**Authors:** Yuanyuan Zhang, Yulin Ding, Junxi Liang, Kai Zhang, Hong Su, Daqing Wang, Min Zhang, Feifei Zhao, Zhiwei Sun, Zhimin Wu, Fenglong Wang, Guifang Cao, Yong Zhang

**Affiliations:** 1College of Veterinary Medicine, Inner Mongolia Agricultural University, Hohhot 010010, China; zyyworkaccount@163.com (Y.Z.); dingyulin2001@126.com (Y.D.); ljx2223128@163.com (J.L.); zhangkai040423@163.com (K.Z.); hongsu1995@126.com (H.S.); wangdaqing050789@126.com (D.W.); zhangmin5400@126.com (M.Z.); imauzff@126.com (F.Z.); sunzhiwei4777@163.com (Z.S.); fengloowang@sohu.com (F.W.); 2Animal Embryo and Developmental Engineering Key Laboratory of Higher Education, Institutions of Inner Mongolia Autonomous Region, Hohhot 010011, China; 3Inner Mongolia Autonomous Region Key Laboratory of Basic Veterinary Medicine, Hohhot 010011, China; 4College of Life Sciences, Inner Mongolia University, Hohhot 010011, China; wuzhimin_21@163.com

**Keywords:** bovine mastitis, dairy mammary epithelial cell, *E. coli*, LPS, MAPK

## Abstract

Bovine mastitis is a condition typically induced by various pathogens, with Escherichia coli (*E. coli*) being a common causative agent known for its propensity to cause persistent infections. In experimental models of bovine mastitis, lipopolysaccharide (LPS), a key component of the *E. coli* cell wall, is frequently employed as an inducer. The extracellular matrix (ECM) is regulated by MMPs, TIMPs, and the uPA system. They collectively participate in ECM degradation and remodeling and have been identified as promising targets for mastitis treatment. However, investigations into the precise mechanisms underlying *E. coli* and LPS-induced mastitis, as well as the relationship between bovine mastitis and the MAPK signaling pathway, remain limited. In this study, bovine mammary epithelial cells (BMECs) were treated in vitro with 10^6^ CFU/mL heat-inactivated *E. coli*, 7.5 µg/mL LPS, or a combination of both. The treatments resulted in varying degrees of activation of the MAPK signaling pathway, specifically ERK1/2, JNK, and P38. BMECs were exposed to MAPK inhibitors (the JNK inhibitor SP600125, the ERK inhibitor PD98059, and the P38 inhibitor SB203580) after treatments with heat-inactivated *E. coli* (10^6^ CFU/mL), LPS (7.5 µg/mL), or a combination of the two for 6, 12, 24, and 48 h. The mRNA and protein levels of *MMP-1*, *MMP-2*, *MMP-3*, *MMP-9*, *MMP-13*, *TIMP-1*, *TIMP-2*, *uPA*, *uPAR*, and *PAI-1* were assessed using RT-qPCR and Western blot analysis. The findings indicated that heat-inactivated *E. coli* and LPS stimulated the expression of MAPK mRNAs *(ERK1/2*, *P38*, *and JNK*) in BMECs, along with corresponding increases in the phosphorylated proteins. Furthermore, MAPK inhibitors substantially upregulated the expression of TIMP-1, TIMP-2, and PAI-1. However, no significant changes were observed in the mRNA and protein levels of *MMP-1*, *MMP-2*, *MMP-3*, *MMP-9*, *MMP-13*, *uPA*, or *uPAR*. In conclusion, heat-inactivated *E. coli* and LPS can activate the MAPK signaling pathway in BMECs. Inhibiting this signaling pathway can modulate the expression of *TIMP-1*, *TIMP -2*, and *PAI-1* at both mRNA and protein levels.

## 1. Introduction

Mastitis can be categorized into several types based on clinical symptoms, including acute mastitis, subacute mastitis, and chronic mastitis. Acute mastitis is typically characterized by sudden clinical symptoms such as fever, redness and swelling of the udder, pain, and a decrease in milk production. In contrast, chronic mastitis may not present with obvious clinical symptoms but can lead to increased somatic cell counts and a higher risk of recurrent infections. This study focuses on chronic mastitis. Bovine mammary fibrosis is characterized by the excessive accumulation of extracellular matrix (ECM) components and the proliferation of myofibroblasts [1]. Escherichia coli (*E. coli*) is a primary pathogen responsible for both clinical and subclinical bovine mastitis [2]. This form of mastitis is particularly challenging to treat and often progresses into chronic conditions in later stages. In many cases, chronic inflammation advances to mammary fibrosis, impairing lactation function. Bovine mammary gland cells (BMECs) produce fibrotic growth factors that promote epithelial–mesenchymal transformation (EMT), thereby increasing ECM production and facilitating fibrosis [3]. These cells also upregulate matrix metalloproteinases (MMPs) and the tissue inhibitors of metalloproteinases (TIMPs), and they activate the urokinase-type plasminogen activator (uPA) system [4].

MMPs, also known as matrixins, are endoproteinases that degrade ECM proteins, playing a crucial role in the restoration and remodeling of the ECM and basement membrane [5]. This regulation is crucial for maintaining ECM integrity during both physiological and pathological processes. Studies have shown that MMP activity fluctuates during the development and resolution of fibrosis [6]. The activation of MMPs is controlled by precursor zymogens and regulated by endogenous inhibitors, i.e., TIMPs [7]. Four TIMPs (TIMP-1, TIMP-2, TIMP-3, and TIMP-4) are expressed in vertebrates and modulated during tissue development and remodeling [8]. Under pathological conditions, where MMP activity becomes dysregulated, changes in TIMP levels play a pivotal role by directly influencing MMP function [9]. Despite extensive investigations into MMPs and TIMPs in the context of fibrosis in organs such as the lung, liver, kidney, and myocardium, their involvement in *E. coli*-induced bovine mammary fibrosis remains unexplored.

Endotoxins, such as lipopolysaccharides (LPS), as major components of the outer membrane of Gram-negative bacteria, trigger MMP activation via a cascade involving plasminogen activators (PAs) [10]. The uPA system comprises uPA, the uPA receptor (uPAR), and its physiologic inhibitor, PA inhibitor-1 (PAI-1) [11]. uPA converts plasminogen into plasmin and signals through uPAR, a process modulated by secreted MMPs [12]. PAI-1 maintains tissue homeostasis by inhibiting MMP activation. Studies of ECM catabolism in fibrosis across other organs suggest similar regulatory mechanisms for MMPs in mammary fibrosis [13].

The MAPK signaling pathway regulates various cellular processes, including proliferation, differentiation, apoptosis, and stress responses. MAPKs, a family of serine/threonine kinases found in eukaryotic cells, phosphorylate target proteins to initiate downstream events [14]. The highly conserved signaling pathway is divided into four subgroups: extracellular regulated kinase (ERK), P38, C-Jun N-terminal kinase (JNK), and ERK5, which form parallel MAPK pathways [15]. In response to various stimuli, distinct MAPK cascades are activated, leading to specific cellular outcomes [16].

This study investigated the role of MMPs, TIMPs, and the uPA system in bovine mammary gland fibrosis, aiming to determine the involvement of the MAPK signaling pathway in *E. coli*-induced and LPS-induced fibrosis and assess the impact of *E. coli* on MMPs, TIMPs, and the uPA system when the MAPK pathway is inhibited in BMECs. This research provides a foundation for deepening the understanding of bovine mammary gland fibrosis.

## 2. Results

### 2.1. The Expression of MAPK Genes in BMECs After Treatments with Heat-Inactivated E. coli and LPS

BMECs were treated with 10^6^ CFU/mL heat-inactivated *E. coli* and 7.5 µg/mL LPS for 6, 12, 24, and 48 h. Total RNA was extracted from the cells. mRNA transcription levels of *ERK1*, *ERK2*, *JNK*, and *P38* were detected by RT-qPCR to assess the activation of the MAPK signaling pathway. The results showed that compared to the control group, the expression of *ERK1* mRNA decreased remarkably after treatments with 10^6^ CFU/mL *E. coli* for 12 h (*p* < 0.05) and with 7.5 µg/mL LPS for 12 h (*p* < 0.05). No significant differences in mRNA levels were observed at other time points (*p* > 0.05) (Figure 1A). The expression of *ERK2* mRNA significantly dropped at 6 h following treatment with 7.5 µg/mL LPS (*p* < 0.001), with no significant differences at other time points (P > 0.05) (Figure 1B). The expression of *JNK* mRNA increased significantly under 10^6^ CFU/mL *E. coli* treatment for 6 h (*p* < 0.001), while it evidently decreased at 6 h following treatment with 7.5 µg/mL LPS (*p* < 0.001). No significant differences were observed at other time points (*p* > 0.05) (Figure 1C). The expression of *P38* mRNA reduced substantially after 6 h of treatment with 7.5 µg/mL LPS (*p* < 0.05). It showed a remarkable increase at 12 h (*p* < 0.001) in the combined *E. coli* + LPS group and no significant changes at other time points (*p* > 0.05) (Figure 1D).

### 2.2. The Expression of MAPK Proteins in BMECs After Treatments with Heat-Inactivated E. coli and LPS

BMECs were treated with 10^6^ CFU/mL heat-inactivated *E. coli* and 7.5 µg/mL LPS for 6, 12, 24, and 48 h to extract cell protein. The expression levels of p-ERK1/2, p-JNK, and p-P38 were detected by Western blot to identify the involvement of the MAPK signaling pathway in MMPs, TIMPs, and the uPA system. The results showed that compared to the control group, p-ERK1/2 expression in the 10^6^ CFU/mL *E. coli* treatment group increased significantly at 6 h (*p* < 0.001) and decreased at 12, 24, and 48 h (*p* < 0.001). In the 7.5 µg/mL LPS treatment group, p-ERK1/2 expression increased significantly at 6 h (*p* < 0.001) and 12 h (*p* < 0.05) and dropped at 48 h (*p* < 0.001). In the *E. coli* + LPS group, p-ERK1/2 expression elevated significantly at 6 h (*p* < 0.001) and diminished at 12, 24, and 48 h (*p* < 0.001) (Figure 2A).

p-JNK expression in the 10^6^ CFU/mL *E. coli* treatment group significantly raised at 6 h (*p* < 0.01) and decreased at 24 and 48 h (*p* < 0.001). It dropped significantly in the 7.5 µg/mL LPS and *E. coli* + LPS groups at 6, 12, 24, and 48 h (*p* < 0.001) (Figure 2B).

p-P38 expression in the 10^6^ CFU/mL *E. coli* treatment group significantly increased at 12 h (*p* < 0.001) and reduced at 6 and 48 h (*p* < 0.001). In the 7.5 µg/mL LPS treatment group, p-P38 expression elevated significantly at 6 and 12 h (*p* < 0.001) and dropped at 24 h (*p* < 0.01) and 48 h (*p* < 0.001). In the *E. coli* + LPS group, p-P38 expression decreased significantly at 6 h (*p* < 0.001), with no significant differences observed at other time points (Figure 2C).

### 2.3. Effects of MAPK Inhibitors on the Expression of MMPs in BMECs

P38-inhibitor + *E. coli* significantly upregulated *MMP-1* mRNA transcription at 24 h. ERK and JNK inhibitors remarkably downregulated *MMP-1* mRNA transcription at 48 h. MAPK inhibitors significantly downregulated the expression of MMP-1 protein at 24 h but had no significant effect at 48 h (Figure 3A).

No evident effect on *MMP*-2 mRNA transcription was observed at either 24 or 48 h. MAPK inhibitors had little effect on the expression of MMP-2 protein at 24 h but significantly upregulated it at 48 h (Figure 3B).

At 24 h, P38-inhibitor + *E. coli* and JNK-inhibitor + LPS both substantially enhanced *MMP-3* mRNA transcription. However, no significant effect on *MMP-3* mRNA transcription was observed at 48 h. MAPK inhibitors barely affected the expression of MMP-3 protein at 24 h. The MAPK inhibitors + LPS group and MAPK inhibitors + *E. coli* + LPS significantly downregulated the expression of MMP-3 protein at 48 h (Figure 3C).

There was no significant effect on *MMP-9* mRNA transcription at 24 h. P38-inhibitor + *E. coli* + LPS markedly upregulated *MMP-9* mRNA transcription at 48 h. MAPK inhibitors evidently downregulated the expression of MMP-9 protein at 24 h. The P38-inhibitor + *E. coli*, P38-inhibitor + LPS, ERK-inhibitor + LPS, and JNK inhibitor + *E. coli* + LPS significantly upregulated MMP-9 protein expression at 48 h (Figure 3D).

JNK-inhibitor + LPS substantially upregulated *MMP-13* mRNA transcription at 24 h, and JNK-inhibitor + *E. coli* + LPS significantly upregulated *MMP-13* mRNA transcription at 48 h. At 24 h, JNK-inhibitor + LPS and ERK-inhibitor + LPS remarkably downregulated MMP-13 protein expression. At 48 h, P38-inhibitor + *E. coli*, JNK-inhibitor + LPS, ERK-inhibitor + LPS, JNK-inhibitor + *E. coli* + LPS, and ERK-inhibitor + *E. coli* + LPS significantly upregulated the expression of MMP-13 protein (Figure 3E).

### 2.4. Effects of MAPK Inhibitors on the Expression of TIMPs in BMECs

At 24 h, P38-inhibitor + *E. coli* significantly upregulated TIMP-1 mRNA transcription. MAPK inhibitors had no significant effect on the transcription of TIMP-1 mRNA at 48 h, but they significantly upregulated the expression level of TIMP-1 protein at 24 h. Additionally, this expression was evidently downregulated by P38-inhibitor + *E. coli* + LPS at 48 h (Figure 4A).

TIMP-2 mRNA transcription was significantly upregulated by P38-inhibitor + *E. coli* and JNK-inhibitor + LPS at 24 h and by ERK-inhibitor + LPS at 48 h. At 24 and 48 h, MAPK inhibitors substantially downregulated the expression of TIMP-2 protein (Figure 4B).

### 2.5. Effects of MAPK Inhibitors on the Expression of the uPA System in BMECs

uPA mRNA transcription presented no apparent variation at 24 or 48 h. At 24 h, the P38 inhibitor markedly downregulated uPA protein expression. ERK-inhibitor + LPS, P38-inhibitor + LPS, ERK-inhibitor + *E. coli* + LPS, JNK-inhibitor + *E. coli* + LPS, and P38-inhibitor + *E. coli* + LPS significantly upregulated uPA protein expression. At 48 h, uPA protein expression was substantially upregulated in all groups except JNK-inhibitor + *E. coli* and ERK-inhibitor + *E. coli* (Figure 5A).

No evident changes were observed in uPAR mRNA transcription at 24 h. However, P38-inhibitor + *E. coli* significantly upregulated uPAR mRNA transcription at 48 h. JNK-inhibitor + *E. coli* + LPS and P38-inhibitor + *E. coli* + LPS remarkably upregulated the expression of uPAR protein at 24 h. The groups excluding JNK-inhibitor + *E. coli* and ERK-inhibitor + *E. coli* profoundly upregulated uPAR protein expression at 48 h (Figure 5B).

PAI-1 mRNA transcription barely changed at 24 h. At 48 h, JNK-inhibitor + LPS, P38-inhibitor + LPS, ERK-inhibitor + *E. coli* + LPS, JNK-inhibitor + *E. coli* + LPS, and P38-inhibitor + *E. coli* + LPS significantly downregulated the transcription of PAI-1 mRNA. At 24 h, PAI-1 protein expression was profoundly downregulated in all groups except P38-inhibitor + LPS and ERK-inhibitor + *E. coli* + LPS. At 48 h, this expression was significantly downregulated in groups excluding JNK-inhibitor + *E. coli* and ERK-inhibitor + *E. coli* + LPS (Figure 5C).

## 3. Discussion

In healthy organisms, MMPs regulate the ECM by degrading its components. However, during inflammatory responses, ECM overexpression often occurs, which induces upregulation of MMPs and TIMPs [17]. *E. coli* can trigger inflammation and even fibrosis in various tissues and organs. Studies have shown that after *E. coli* infects the villous membrane, there is an increase in the secretion of active forms of MMP-2 and MMP-9, which are related to the ECM, while the levels of TIMP-1, -2, and -4 remain unchanged [18]. MMP-9 activity is essential in reducing cell death triggered by contact-dependent phagocytosis in neonatal monocytes infected with *E. coli*. Through this mechanism, MMP-9 may assist in mitigating persistent inflammation in neonates [19]. Rapamycin likely alleviates the progression of sepsis-induced pulmonary fibrosis by downregulating the expression of TGF-β1, α-SMA, MMP-2, and TIMP-1, thus protecting the lung.

LPS possesses a complicated molecular structure. Located on the peptidoglycan (mucopeptide, MP) layer of the outer membrane of the Gram-negative bacteria, LPS represents the outermost component of the bacterial cell wall. It has both antigenicity and toxicity. LPS can significantly upregulate the expression of fibrosis-related factors, such as FGF-2, uPA, MMP-2, and MMP-9, in primary cardiac fibroblasts. In cells treated with LPS from Porphyromonas gingivalis (1690) and *E. coli*, there is a marked upregulation of MMP-3 mRNA and protein expression. However, in *P. gingivalis* LPS-treated cells (1435/1449), MMP-3 mRNA and protein levels show no significant upregulation [20]. LPS induces osteoblast-like MC3T3-E1 cells to mimic the pathological characteristics of periodontitis. The results indicate that NF-κB significantly enhances the expression of MMP-13. Inhibition of NF-κB profoundly suppresses LPS-induced MMP-13 expression, while C/EBPβ downregulation does not affect the expression of MMP-13. This suggests that LPS can induce the expression of MMP-13 [21] and trigger inflammation and even fibrosis in tissues and organs.

In a rat lung injury model induced by *E. coli* [22], both the mRNA and protein expression of MMP-9 were found to be upregulated. Similarly, in a mouse lung injury model induced by LPS [22], the mRNA and protein levels of MMP-2 and -9 also increased. These findings demonstrate that both *E. coli* and LPS can induce lung injury and promote the expression of MMP-2 and -9. MMP-2 and MMP-9 can degrade type IV collagen in the basement membrane, thereby compromising the structure integrity of lung tissues. In transgenic mouse lung tissue, the expression of TIMP-1, -2, and -3 was found to be upregulated, alongside a reduction in the activity of MMP-9. This reduction may decrease ECM degradation and lead to pulmonary fibrosis [23]. It has been reported [24] that MMP-13 is induced by pro-inflammatory cytokines, with its increased expression linked to the pathogenesis of diseases such as cancer, osteoarthritis, rheumatoid arthritis, and periodontal disease. In mouse periodontal ligament fibroblasts, the LPS of *E. coli* and actinomyces enhanced the level of MMP-13 mRNA by 27% and 46%, respectively. Inhibition of p38 MAP kinase can substantially reduce the expression of MMP-13 mRNA. MMP-2 and MMP-9 serve anti-fibrotic roles in the early stages of inflammation by degrading excess ECM and retarding fibrosis progression. Their functions shift in the later stages, acting as pro-inflammatory mediators, promoting cell proliferation, and initiating and exacerbating the process of fibrosis [25].

The antigenic components of *E. coli* can be categorized into somatic antigens, flagellar antigens, and surface antigens. Surface antigens can resist phagocytosis and complement-mediated killing [26]. LPS, a major constituent of the cell wall of *E. coli*, comprises lipid A, core polysaccharides, and specific polysaccharides. It is a key virulence factor of Gram-negative bacteria. Upon interaction with host cells, it exhibits a variety of biological activities [27]. LPS can provoke inflammatory responses in animals through cellular signal transduction pathways. After binding to lipopolysaccharide-binding protein (LBP) and CD14, LPS forms an LPS-LBP-CD14 trimeric complex that is recognized by the pattern recognition receptor TLR4. This recognition activates the NF-κB signaling pathway and initiates an inflammatory response [28]. LPS-mediated intracellular signal activation involves adaptor molecules, such as myeloid differentiation factor 88 (MyD88), IL-1 receptor-associated kinase 1 (IRAK-1), and IRAK-4. These molecules assemble at the receptor complex via MyD88 and MyD88 adaptor-like protein, leading to the phosphorylation of IRAK-1. Subsequently, IRAK-1 dissociates from the complex and transmits the signal to TRAF-6, thereby activating it.

Due to the complicated pathogenesis of fibrosis, activated signaling pathways vary with tissues and organs. Despite a basic understanding, the relationship between the common signaling pathways remains poorly defined. The MAPK/ERK signaling pathway plays a key role in regulating cellular processes related to fibrosis, including cell growth and proliferation [29]. In contrast, the PAK/P38 signaling pathway primarily participates in the EMT during the pro-fibrotic process and is activated in organ fibrosis [30]. The JNK signaling pathway, activated by various cellular stress responses, is pivotal in cell death and inflammation. Its activation is a hallmark of kidney damage. The application of JNK inhibitors has been shown to prevent acute kidney injury and inhibit the progression of glomerulosclerosis and tubulointerstitial fibrosis. The JNK signaling pathway promotes the production of pro-inflammatory and pro-fibrotic molecules by tubular epithelial cells and facilitates the dedifferentiation of these cells into an interstitial phenotype [31]. In NIH3T3 cells, pretreatment with the ERK1/2-specific inhibitor PD98059 completely blocks TGF-βCM-induced fibroblast proliferation and differentiation, indicating that FGF-2 secreted by alveolar epithelial cells responds to TGF-β1 and induces fibroblast proliferation and fibrosis activation via the ERK kinase pathway [32]. In periostin-deficient mice, the expression of fibrosis/apoptosis markers and phosphorylated P38 MAPK reduced, and inhibiting P38 MAPK improved hypoxia-induced fibrosis. The addition of P38 MAPK inhibitors to recombinant periostin can evidently ameliorate periostin-induced fibrosis [33]. Inhibition of the ERK1/2 pathway has a protective effect against LPS-induced myocardial fibrosis, with a notable reduction in the expression of MMP-2 and MMP-9 [34]. The results demonstrate that heat-inactivated *E. coli* and LPS significantly upregulate the phosphorylation of ERK1/2 and P38 and downregulate that of JNK. After treatments with heat-inactivated *E. coli* and LPS, the MAPK inhibitors marked reduced the protein content of MMP-1 (24 h), MMP-3 (48 h), MMP-9 (24 h), MMP-13 (24 h), TIMP-2 (24 and 48 h), uPA (24 h), and PAI-1 (24 and 48 h). However, it did not affect the protein expression of MMP-2, TIMP-1, and uPAR. The MAPK inhibitors downregulated the expression of uPA in bovine mammary fibroblasts induced by heat-inactivated S. aureus. Meanwhile, inhibiting multiple signal transduction molecules downregulated PAI-1 expression in these cells [35]. This study used three types of MAPK inhibitors on *E. coli* and LPS-induced bovine mammary epithelial cells. The lack of effect on MMP-2, TIMP-1, and uPAR may be attributed to the combined actions of the inhibitors. The results indicate that fibrosis in bovine mammary glands activates the MAPK signaling pathway, providing insights into the potential role of MMPs, TIMPs, and the uPA system in modulating ECM metabolism.

Our study aims to initially reveal the potential roles of certain key genes and proteins in mammary damage. By examining changes in mRNA and protein expression levels, we can preliminarily establish the correlation between the activity changes in these molecules and mammary damage. Due to limitations in experimental conditions and resources, we are currently only able to conduct these basic molecular biology experiments. However, we also recognize that these data alone are insufficient to fully elucidate the complex cellular and molecular mechanisms. Relying solely on mRNA and protein expression analysis to explore the mechanisms of mammary damage has certain limitations. To fully elucidate the complex cellular and molecular mechanisms of mastitis, further relevant studies are needed. The activation of MAPK signaling pathway-related factors and changes in the expression of proteins such as MMPs and TIMPs hold significant diagnostic and prognostic potential in various diseases. Abnormal activation of the MAPK pathway (e.g., ERK, JNK, and p38) is closely associated with cell proliferation, apoptosis, and inflammatory responses, and its phosphorylation levels can serve as markers of disease status. MMPs (e.g., MMP-2, MMP-9) are involved in extracellular matrix degradation, with their overexpression linked to tumor invasion and metastasis, while TIMPs (e.g., TIMP-1, TIMP-2), as natural inhibitors of MMPs, may exacerbate disease progression when their expression is imbalanced. By detecting the activation status of the MAPK pathway, the expression levels of MMPs and TIMPs, and their ratios, disease progression, and prognosis can be assessed. Combined analysis of these molecular markers not only enhances diagnostic sensitivity and specificity but also provides a basis for personalized treatment, demonstrating significant clinical application prospects, particularly in cancers, inflammatory diseases, and fibrosis.

## 4. Materials and Methods

### 4.1. Preparation of Heat-Inactivated E. coli

First, *E. coli* was isolated and identified in the laboratory. The bacterial strain was initially cultured in nutrient broth at 37 °C for 24 h, then transferred to an agar plate for an additional 24 h of cultivation at the same temperature, followed by 18 h amplification. For subsequent experiments, the bacterial concentration was adjusted to 10^6^ CFU/mL and heat-inactivated at 80 °C for 40 min. Then, the *E. coli* was placed on an agar plate and incubated overnight at 37 °C. The final observation showed no colony formation.

### 4.2. Determining the Optimal Dose of LPS

The CCK-8 assay was used following the instructions. BMECs in the logarithmic growth phase were harvested and seeded into 96-well plates with 1 × 10^4^ cells in each well. The cells were incubated at 37 °C for 24 h. Four experimental groups (the concentrations of LPS are 2.5, 5.0, 7.5, and 10.0 µg/L, respectively) were established. Each group had three replicates. Meanwhile, a blank control group was set up, containing only basal medium. After six hours of LPS stimulation, 10 µL of CCK-8 solution was added to each well. The incubation continued for four hours. The absorbance at D600 nm was measured using a microplate reader. The optimal concentration of LPS was determined based on the criterion that the cell inhibition rate (IR) should be ≤10%. The cell inhibition rate was calculated as follows:IR = (LPS stimulated group D600 nm − blank group D600 nm)/(control group D600 nm − blank group D600 nm) × 100%.

The optimal LPS dose for stimulating BMECs was determined to be 7.5 µg/mL.

### 4.3. Cell Treatment

The BMECs used in this experiment were isolated and cryopreserved in our laboratory in advance. The cells were cultured in a DMEM/F12 medium supplemented with 10% fetal bovine serum (FBS; ExCell Biolog, Shanghai, China) and 1% penicillin/streptomycin solution in 25 cm^3^ flasks until 80–90% confluence was reached. Before stimulation, the cells were removed from the culture medium and washed three times with PBS. Subsequently, these cells were incubated for 24 h in a fresh DMEM/F12 medium, supplemented with 1% penicillin/streptomycin solution. Following this, the cells were stimulated with serum-free medium (control group), 10^6^ CFU/mL heat-inactivated *E. coli*, or 7.5 µg/mL LPS (treatment group) for 6, 12, 24, and 48 h, respectively. After stimulation, the supernatant was collected and stored at −80 °C for further analysis. Finally, BMECs were harvested for quantitative PCR (qPCR) and Western blot analysis. All animals used in this study were privately owned by Inner Mongolia Agricultural University. A MAPK signaling pathway inhibitor with a concentration of 20 μM was prepared using serum-free DMEM/F12 medium(ERK inhibitor PD98059, JNK inhibitor SP600125, P38 inhibitor SB203580), stored at −20 °C. When BMECs grew to the 7th generation and filled the cell vials, the cells were starved with FBS-free DMEM/F12 medium for 24 h. In the blank control group, a new FBS-free culture medium was replaced in the cell bottle. In the test group, 1 mL of a 10⁶ CFU/mL heat-inactivated *E. coli* bacterial solution was added. For the other groups, 20 μM MAPK inhibitor was added to BMECs alone after 1 h. After that, the heat-inactivated E. coli solution and LPS were added for 24 and 48 h. Finally, total RNA and total protein were extracted from the cells of each group.

### 4.4. Total RNA Extraction and RT-qPCR Analysis

Total RNA was extracted from BMECs using the AXYGEN kit (Central Avenue Union City, CA, USA), following the manufacturer’s protocol. Reverse transcription was performed using the Prime Script RT Reagent Kit (TaKaRa BIO Inc., San Jose, CA, USA). The conditions for reverse transcription included incubation at 37 °C for 15 min, followed by 85 °C for 5 s. The microplate reader was used to measure the OD values (A260/A280) and concentration of RNA. When the RNA concentration is between 500 and 1000 ng/μL, and the OD value (A260/A280) is between 1.8 and 2.0, both the concentration and purity are considered to be within the acceptable range. The resulting cDNA samples were stored at −80 °C. β-actin expression was used as the endogenous control gene. The qPCR cycling included 50 °C, 2 min; 95 °C, 10 min; 95 °C, 15 s; 60 °C, 60 s. The reaction system was constructed according to the TB^®^Green protocol (TaKaRa BIO Inc.). β-actin, MMP-1, MMP-2, MMP-3, MMP-9, MMP-13, TIMP-1, TIMP-2, uPA, uPAR, PAI-1, ERK1, ERK2, JNK, P38 primers were designed and synthesized by Shengong Biological Co., Ltd. (Shanghai, China). The specific primer sequence is presented in Table 1.

### 4.5. Western Blot Analysis

The samples were collected following the instructions provided in the total protein extraction kit for tissue or cells (Sangon Biotech, Shanghai, China). Protein concentrations were gauged using the BCA protein assay kit (Beyotime Biotechnology, Shanghai, China). After denaturation at 100 °C for 5 min, the samples were separated by electrophoresis on an 8–10% SDS-PAGE gradient gel. The corresponding gel region was excised and transferred to a suitably sized nitrocellulose (NC) membrane. Then, the membrane was blocked with 5% bovine serum albumin (BSA) at room temperature for 4 h. Primary antibody incubation was carried out for 12 h at 4 °C with the following antibodies: mouse-derived anti-β-actin (Abcam, Cambridge, UK; diluted 1:10,000), rabbit-derived anti-MMP-1 (Bioworld Technology, Nanjing, China; diluted 1:300), and rabbit-derived anti-MMP-1, -2, -3, -9, -13, TIMP-1, -2, uPA, uPAR, PAI-1, ERK1/2, P38, JNK (ProteinTech Group, San Diego, CA, USA; diluted 1:500). After washing with TBST for 5 to 10 min, the membrane was incubated with secondary antibodies (goat anti-rabbit, diluted 1:3000, and goat anti-mouse, diluted 1:4000) for 1 h. Immunoblot signals were detected with the ECL Western blot detection kit (Thermo Scientific, Waltham, MA, USA) according to the manufacturer’s instructions. As the PVDF membrane was cut during the transfer, only the regions containing bands were retained for analysis.

### 4.6. Statistical Analysis

mRNA expression levels were normalized to β-actin using the 2^−ΔΔCT^ method [36]. SPSS 22 was employed for statistical analysis. Western blot band quantification was performed using ImageJ software, version 1.48V and GraphPad Prism 8. The results were presented as means ± SD, with statistical significance determined by two-way ANOVA (* *p* < 0.05, ** *p* < 0.01, *** *p* < 0.001).

### 4.7. Materials and Reagents

LPS was purchased from InvivoGen (San Diego, CA, USA), BSA from Amresco (Solon, OH, USA), fetal bovine serum from ExCell Biolog (Shanghai, China), DMEM/F12 medium, 0.25% pancreatic enzyme, and double antibody from Gibco (Waltham, MA, USA). The Multi-source Total RNA Miniprep Kit was purchased from AXYGEN (Union City, CA, USA), the total protein extraction kit from Shengong Bioengineering (Shanghai, China) Co., Ltd., and the reverse transcription kit and qPCR kit from TaKaRa (San Jose, CA, USA). BCA protein concentration determination kit and 5× loading buffer were purchased from Beyotime (Shanghai, China). Protein predye Maker was purchased from Bio-Rad (Hercules, CA, USA). SDS-PAGE gel preparation kit was purchased from Solarbio (Beijing, China). MMP-1, MMP-2, MMP-3, MMP-9, MMP-13, TIMP-1, TIMP-2, uPA, uPAR, PAI-1 rabbit MAB were purchased from Proteintech (San Diego, CA, USA). Goat anti-rabbit IgG-HRP, goat anti-mouse IgG-HRP and mouse anti-β-Actin monoclonal antibodies were purchased from Tianjin Sanjian Biotechnology Co., Ltd. (Tianjing, China) MMP-2 and MMP-9 gelatin enzyme assay kits were purchased from Shanghai Xinfan Biotechnology Co., Ltd. (Shanghai, China).

### 4.8. Laboratory Apparatus

The 37 °C constant temperature biochemical incubator (HCP108) was purchased from Merter (Dreieich, Germany). The High-Speed refrigerated centrifuge (X3R) was purchased from Thermo Fisher (Waltham, MA, USA). The inverted microscope (TS100) was purchased from Nikon (Tokyo, Japan). The Elix Pure Water instrument (CDUFB1001) was purchased from Millipore (Burlington, MA, USA). The Biosafety cabinet (BSC-1300A2) was purchased from Suzhou Zhijing Purification Instrument Co., Ltd. (Suzhou, China) The multifunctional enzyme marker (Synergy H5) was purchased from BioTek Company (Shoreline, WA, USA). The PCR instrument (2720) and RT-qPCR instrument (VIIA7) were purchased from ABI Company (Los Angeles, CA, USA). The Vortex mixer (D1008E) was purchased from Celor Czech (Kentwood, MI, USA). The Power-Pac electrophoresis apparatus (1658033) and the semi-dry transfer apparatus Trans-Blot^®^TurboTM were purchased from Bio-Rad (Hercules, CA, USA).

## 5. Conclusions

Heat-inactivated *E. coli* and LPS activate factors related to the MAPK signaling pathway, promoting the phosphorylation levels of ERK1/2, P38, and JNK in BMECs. Inhibitors of the MAPK pathway modulate the expression of MMP-1, MMP-3, MMP-9, MMP-13, TIMP-2, uPA, and PAI-1. This mechanism provides new insights into the treatment of inflammation-related diseases. In terms of regulating inflammatory responses, inhibiting the MAPK signaling pathway can effectively control the production of inflammatory factors, especially in LPS-induced inflammatory models, where this regulatory effect has been extensively studied. Additionally, using cell models induced by heat-inactivated *E. coli* and LPS, it is possible to screen for novel compounds with anti-inflammatory potential and provide a reliable experimental platform for drug development. In clinical applications, by studying the expression differences in the MAPK pathway in different individuals, it is expected to develop personalized anti-inflammatory treatment plans. Moreover, combining MAPK inhibitors with other anti-inflammatory drugs can lead to more effective combination therapies. Future research will further explore the specific mechanisms of this pathway under various inflammatory conditions, offering broader prospects for the treatment of inflammation-related diseases, drug development, and the construction of disease models.

## Figures and Tables

**Figure 1 ijms-26-03893-f001:**
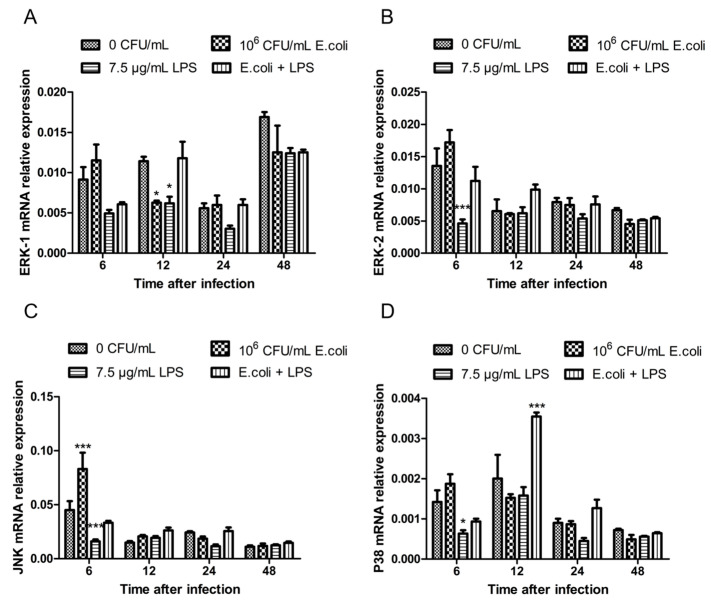
The relative expression levels of MAPK mRNA. (**A**) Expression of ERK1 mRNA in BMECs after treatment with heat-inactivated *E. coli* and LPS for 6, 12, 24, and 48 h. (**B**) Expression of ERK2 mRNA in BMECs after treatment with heat-inactivated *E. coli* and LPS for 6, 12, 24, and 48 h. (**C**) Expression of JNK mRNA in BMECs after treatment with heat-inactivated *E. coli* and LPS for 6, 12, 24, and 48 h. (**D**) Expression of P38 mRNA in BMECs after treatment with heat-inactivated *E. coli* and LPS for 6, 12, 24, and 48 h. * *p* < 0.05; *** *p* < 0.001.

**Figure 2 ijms-26-03893-f002:**
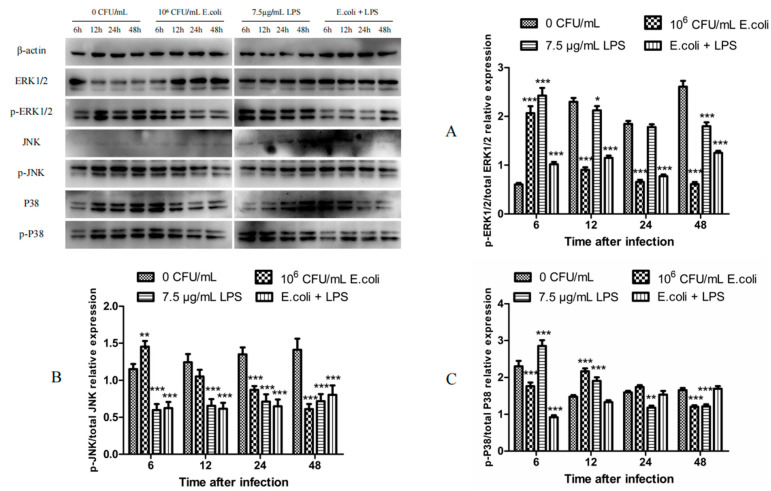
Relative expression levels of MAPK proteins. (**A**) Western blot results of the relative expression of ERK1/2 (42/44 kDa) and phosphorylated ERK1/2 (42/44 kDa) proteins after treatments with heat-inactivated *E. coli* and LPS for 6, 12, 24, and 48 h. (**B**) Western blot results of the relative expression of JNK (46 kDa) and phosphorylated JNK (p-JNK, 46 kDa) proteins in BMECs after treatment with heat-inactivated *E. coli* and LPS for 6, 12, 24, and 48 h. (**C**) Western blot results of the relative expression of P38 (43 kDa) and phosphorylated P38 (p-P38, 43 kDa) proteins in BMECs after treatment with heat-inactivated *E. coli* and LPS for 6, 12, 24, and 48 h. * *p* < 0.05; ** *p* < 0.01; *** *p* < 0.001.

**Figure 3 ijms-26-03893-f003:**
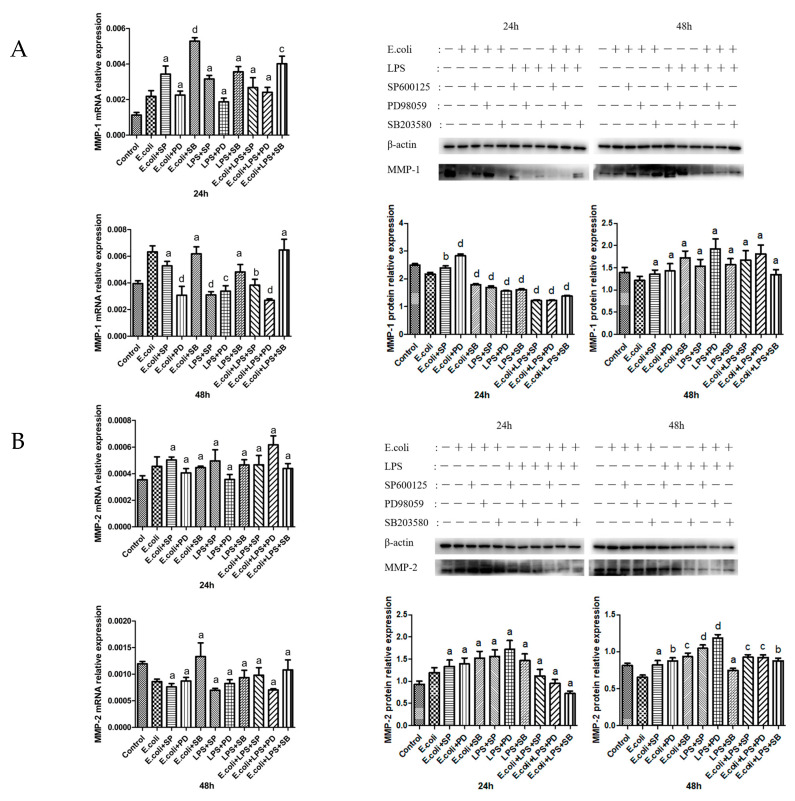
The expression of MMP mRNA and protein after adding MAPK inhibitors. (**A**) The expression of MMP-1 mRNA and protein (54 kDa) in BMECs after adding MAPK inhibitors for 24 and 48 h. (**B**) The expression of MMP-2 mRNA and protein (62 kDa) in BMECs after adding MAPK inhibitors for 24 and 48 h. (**C**) The expression of MMP-3 mRNA and protein (60 kDa) in BMECs after adding MAPK inhibitors for 24 and 48 h. (**D**) The expression of MMP-9 mRNA and protein (67 kDa) in BMECs after adding MAPK inhibitors for 24 and 48 h. (**E**) The expression of MMP-13 mRNA and protein (70 kDa) in BMECs after adding MAPK inhibitors for 24 and 48 h. SP represents the JNK inhibitor SP600125, PD denotes the ERK inhibitor PD98059, and SB refers to the P38 inhibitor SB203580; a, b, c, and d signify significant differences compared to the *E. coli* group, with “a” indicates that *p* > 0.05, “b” indicates that *p* < 0.05, “c” indicates that *p* < 0.01, “d” indicates that *p* < 0.001, respectively.

**Figure 4 ijms-26-03893-f004:**
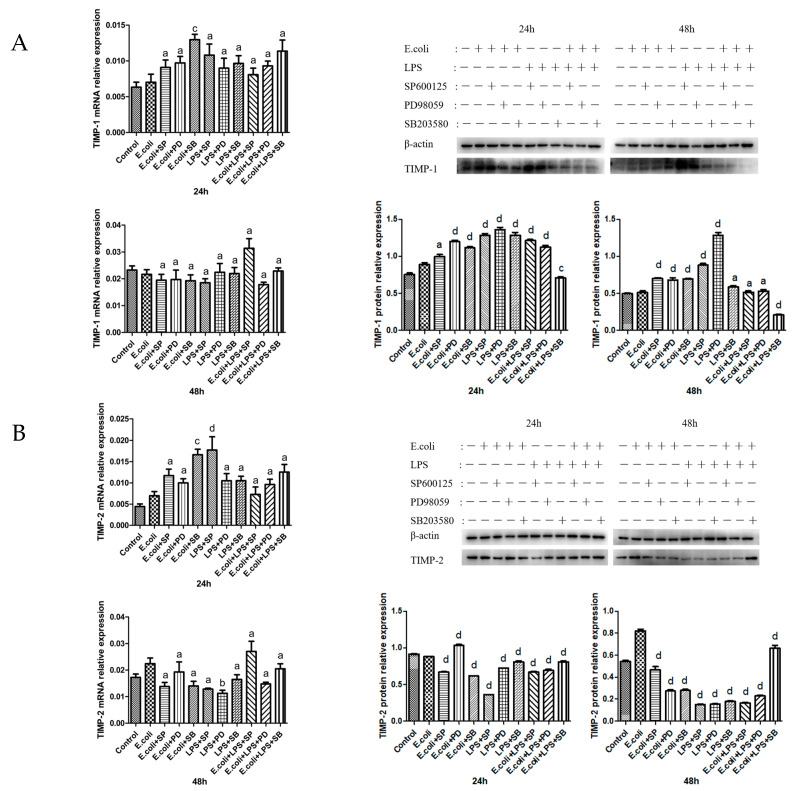
The expression of TIMP mRNA and protein following the addition of MAPK inhibitors. (**A**) The expression of TIMP-1 mRNA and protein (23 kDa) following the addition of MAPK inhibitors in BMECs for 24 and 48 h. (**B**) The expression of TIMP-2 mRNA and protein (28 kDa) following the addition of MAPK inhibitors in BMECs for 24 and 48 h. SP denotes the JNK inhibitor SP600125, PD refers to the ERK inhibitor PD98059, and SB represents the P38 inhibitor SB203580; a, b, c, and d indicate significant differences compared to the *E. coli* group, with “a” indicates that *p* > 0.05, “b” indicates that *p* < 0.05, “c” indicates that *p* < 0.01, “d” indicates that *p* < 0.001, respectively.

**Figure 5 ijms-26-03893-f005:**
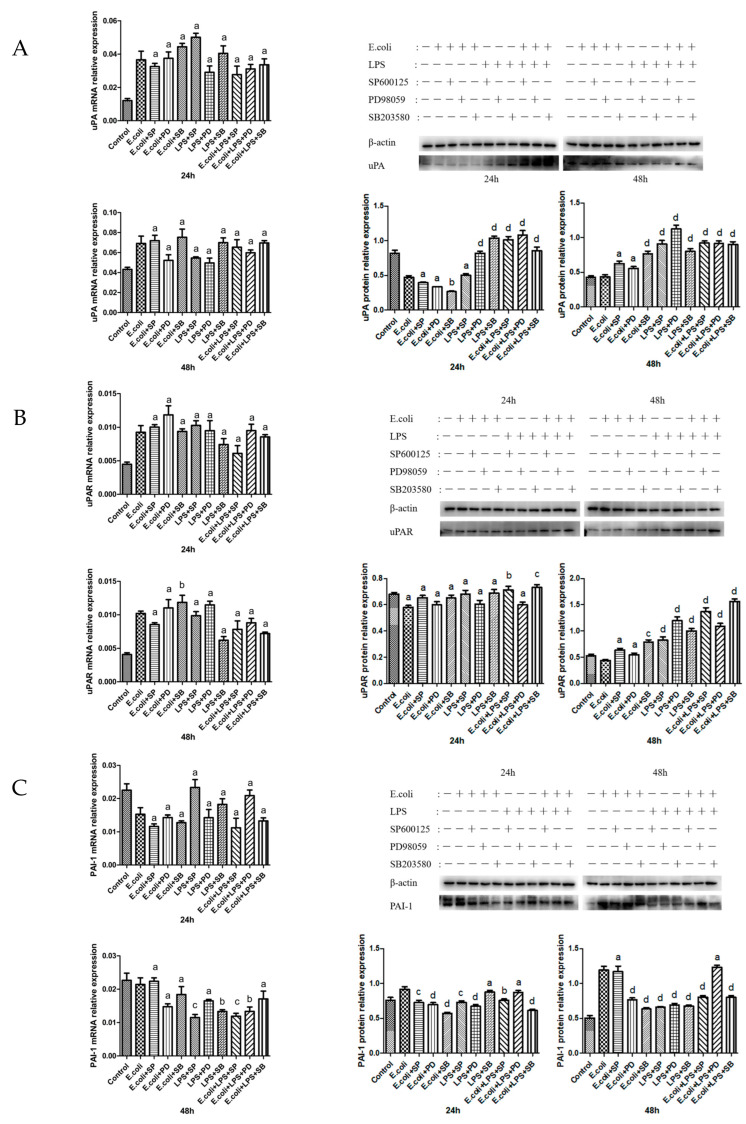
The expression of uPA mRNA and protein after adding MAPK inhibitors. (**A**) The expression of uPA mRNA and protein (54 kDa) after adding MAPK inhibitors in BMECs for 24 and 48 h. (**B**) The expression of uPAR mRNA and protein (37 kDa) after adding MAPK inhibitors in BMECs for 24 and 48 h. (**C**) The expression of PAI-1 mRNA and protein (45 kDa) after adding MAPK inhibitors in BMECs for 24 and 48 h. SP refers to the JNK inhibitor SP600125, PD signifies the ERK inhibitor PD98059, and SB denotes the P38 inhibitor SB203580; a, b, c, and d represent significant differences compared to the *E. coli* group, with “a” indicates that *p* > 0.05, “b” indicates that *p* < 0.05, “c” indicates that *p* < 0.01, “d” indicates that *p* < 0.001, respectively.

**Table 1 ijms-26-03893-t001:** Primer sequences for real-time PCR.

Gene Name	Primer Sequence (5′–3′)	TM (°C)	Product Size/bp
*MMP-1*	F: TCAACTCTGGAGCAATGTCACACC	59.2	175
	R: ATGAGCGTCTCCTCCGATACCTG		
*MMP-2*	F: GACCAGAGCACCATTGAGACCATG	58.9	169
	R: GAGCGAAGGCATCATCCACTGTC		
*MMP-3*	F: AGAGTCTTCCGATTCTGCTGTTGC	58.5	93
	R: GCTCCATGGTGTCTTCCTTGTCC		
*MMP-9*	F: GGTGCTGGCTTGCTGCTCTG	56.7	87
	R: TTGGTGAGGTTGGTTCGTGGTTC		
*MMP-13*	F: CATCCTCAGCAGGTTGAAGCAGAG	58.2	83
	R: TCATAGGCGGCATCAATACGGTTG		
*TIMP-1*	F: CCTGTTGCTGCTGTGGCTCAC	58.1	91
	R: GACGACATCGGAGTTGCAGAAGG		
*TIMP-2*	F: ACGAGTGCCTCTGGATGGACTG	56.6	91
	R: GAGCCGTCGCTTCTCTTGATGC		
*uPA*	F: CAGGTCACCAACGCCGAGAAC	53.8	98
	R: CTGATGAGGCTGCCACCACAC		
*uPAR*	F: GCCAACCGCTGCTGCTACTG	58.6	175
	R: ACGTTCATCTCATTGCCACCTTCC		
*PAI-1*	F: GAGAGCCAGGTTCATCGTCAACG	57.9	199
	R: GGTGCTGCCATCGGACTTGTG		
*β-actin*	F: TCTGGCACCACACCTTCTACAAC	60.1	170
	R: GGACAGCACAGCCTGGAT		
*ERK1*	F: ACGTCATTGGCATCCGAGACATTC	57.6	192
	R: GCACGTTGGCGGAGTGGATATAC		
*ERK2*	F: AACCTTCCAACCTGCTGCTCAAC	57.7	182
	R: GATGTCGATGGACTTGGTGTAGCC		
*JNK*	F: ATCTGGTCTGTTGGCTGTATAATGGC	57.3	132
	R: CCTGGACATGAAGTCTTGGCTTGG		
*P38*	F: GGCTCCTGAGATCATGCTGAACTG	57.5	137
	R:TCTGCTGAAGCTGGTTAATATGGTCTG		

## Data Availability

Data and materials are presented in the paper.

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
