# Peer review of "Impact of Escherichia coli and Lipopolysaccharide on the MAPK Signaling Pathway, MMPs, TIMPs, and the uPA System in Bovine Mammary Epithelial Cells"

_ijms, 2025, doi:10.3390/ijms26083893_

Round 1

Reviewer 1 Report

Comments and Suggestions for Authors

The study investigates the mechanisms of mammary gland damage induced by E. coli and LPS using bovine mammary epithelial cells, which is a relevant and important topic in the field. However, the manuscript has several critical issues that need to be addressed:

  1. The study relies solely on mRNA and protein expression analysis via Western blot. This approach is insufficient to draw definitive conclusions regarding the mechanisms of mammary damage. 
  2. The repeated description that bovine mastitis is a “chronic inflammation” is inaccurate. Mastitis can be acute or chronic, and this distinction is essential for understanding the disease. The authors should correct this terminology to reflect the complexity of mastitis and specify the type of inflammation relevant to their study.
  3. The manuscript contains numerous formatting errors that detract from its readability. For example, E. coli should be italicized, and statistical significance should be formatted as P < 0.05. Additionally, proper spacing between numbers and units is lacking (e.g., “Fig4” should be “Fig. 4,” and “240-241” should be “240–241”). These issues are pervasive and need to be corrected throughout the text.
Comments on the Quality of English Language

The quality of the English is average, but there are many cases where the formatting is not standardized, which affects the readability.

Author Response

Comments 1: The study relies solely on mRNA and protein expression analysis via Western blot. This approach is insufficient to draw definitive conclusions regarding the mechanisms of mammary damage.

Response 1: Thank you very much for your attention to our study and your valuable comments. Your remarks on the limitations of our research methods are very pertinent. We fully understand the potential shortcomings of relying solely on mRNA and protein expression analysis to explore the mechanisms of mammary damage. In our study, we chose mRNA and protein expression analysis as the primary means for the following reasons. Our study aims to initially reveal the potential roles of certain key genes and proteins in mammary damage. By examining changes in mRNA and protein expression levels, we can preliminarily establish the correlation between the activity changes of these molecules and mammary damage. Due to limitations in experimental conditions and resources, we are currently only able to conduct these basic molecular biology experiments. However, we also recognize that these data alone are insufficient to fully elucidate the complex cellular and molecular mechanisms. We will supplement the discussion section of our paper with an explanation of the limitations of these methods and emphasize the necessity for future research. Thank you for your valuable feedback. We will use this opportunity to further improve our study.

Comments 2: The repeated description that bovine mastitis is a “chronic inflammation” is inaccurate. Mastitis can be acute or chronic, and this distinction is essential for understanding the disease. The authors should correct this terminology to reflect the complexity of mastitis and specify the type of inflammation relevant to their study.

Response 2: Thank you for your suggestions. Your remarks on the inaccuracy of describing bovine mastitis as “chronic inflammation” are very pertinent. You are absolutely right that mastitis can be either acute or chronic, and this distinction is essential for understanding the disease. In our study, we indeed did not explicitly differentiate between the types of mastitis, which could lead to misunderstandings. In fact, mastitis can be categorized into several types based on clinical symptoms, including acute mastitis, subacute mastitis, and chronic mastitis. Acute mastitis is typically characterized by sudden clinical symptoms such as fever, redness and swelling of the udder, pain, and a decrease in milk production. In contrast, chronic mastitis may not present with obvious clinical symptoms but can lead to increased somatic cell counts and a higher risk of recurrent infections. To more accurately reflect the complexity of mastitis, we will explicitly specify the types of mastitis in our manuscript and clearly state the specific type of inflammation relevant to our study based on our research subjects and experimental design. Since our study primarily focuses on chronic mastitis, we will explicitly state in the text, “This study focuses on chronic mastitis,” and provide additional background information accordingly(Page 2, Line 43-48). We will take this opportunity to further enhance the rigor of our research and presentation.

Comments 3: The manuscript contains numerous formatting errors that detract from its readability. For example, E. coli should be italicized, and statistical significance should be formatted as P < 0.05. Additionally, proper spacing between numbers and units is lacking (e.g., “Fig4” should be “Fig. 4,” and “240-241” should be “240–241”). These issues are pervasive and need to be corrected throughout the text.

Response 3: Thank you for your valuable comments. We fully understand the importance of formatting standards for the readability and professionalism of a paper, and we are also aware that formatting errors can interfere with the reader's comprehension. We sincerely apologize for these issues you have pointed out and will immediately take the following measures to correct them. We will ensure that all biological names are italicized to comply with the norms of scientific writing. We will standardize all notations of statistical significance to ensure they meet academic standards. We will carefully review and correct spacing issues between numbers and units, to ensure the correctness and consistency of the format. We will conduct a thorough review of the entire manuscript to identify and correct all similar formatting issues, thereby enhancing the overall readability and professionalism of the paper(Page 9, Line 258,262). Thank you for your patience and suggestions. We place great importance on your feedback and will use this opportunity to further improve our manuscript. If you have any additional comments or suggestions, we are always open to them and will handle them with care.

Reviewer 2 Report

Comments and Suggestions for Authors

The manuscript exhibits significant deficiencies in quality. Numerous shortcomings are evident, particularly in the Materials and Methods section, where the descriptions of measurements lack precision. Critical information is absent, including but not limited to the number of experimental animals, number of technical replications, the specific names and sequences of primers along with their melting temperatures (TM), the identification of apparatus employed, the employment of inhibitors, and a description of certain treatments (LPS+E.coli). These inadequacies complicate the interpretation of the results. Therefore, in my assessment, the manuscript is unacceptable for publication.

Author Response

Comments 1: The manuscript exhibits significant deficiencies in quality. Numerous shortcomings are evident, particularly in the Materials and Methods section, where the descriptions of measurements lack precision. Critical information is absent, including but not limited to the number of experimental animals, number of technical replications, the specific names and sequences of primers along with their melting temperatures (TM), the identification of apparatus employed, the employment of inhibitors, and a description of certain treatments (LPS+E.coli). These inadequacies complicate the interpretation of the results. Therefore, in my assessment, the manuscript is unacceptable for publication.

Response 1: We sincerely thank the reviewer for their thorough and constructive feedback, which has helped us significantly improve the manuscript. In response to the concerns raised, we have revised the Materials and Methods section to include precise details, such as the number of experimental animals, technical replications, primer sequences with melting temperatures, apparatus identification, inhibitor usage, and a clear description of treatments (e.g., LPS + E. coli). Additionally, we have enhanced the overall quality of the manuscript by improving clarity, consistency, and transparency. We believe these revisions address all the issues raised and hope the revised version meets the journal's standards for publication. Thank you for your valuable input. (Changes can be found-Page 3, Line 129-138; Page 4, Line 147-152; Page 5, Line 177-200).

Reviewer 3 Report

Comments and Suggestions for Authors

The authores demonstrated how heat-inactivated E. coli and LPS activate factors related to the MAPK signaling pathway, promoting the phosphorylation levels of ERK1/2, P38, and JNK in BMECs. Inhibitors of the MAPK pathway modulate the expression of MMP-1, MMP-3, MMP-9, MMP-13, TIMP-2, uPA, and PAI-1.

I recommend to improve discussion e further future persepctive to how to use these information in a diagnostic and applied fiel in the discussion and cocnclusions.

Please check all the Italics, when needed.

Comments on the Quality of English Language

the quality can be improved

Author Response

Comments 1: The authores demonstrated how heat-inactivated E. coli and LPS activate factors related to the MAPK signaling pathway, promoting the phosphorylation levels of ERK1/2, P38, and JNK in BMECs. Inhibitors of the MAPK pathway modulate the expression of MMP-1, MMP-3, MMP-9, MMP-13, TIMP-2, uPA, and PAI-1.

I recommend to improve discussion e further future persepctive to how to use these information in a diagnostic and applied fiel in the discussion and cocnclusions.

Please check all the Italics, when needed.

Response 1: Thank you very much for your attention to our study and your valuable suggestions. We highly appreciate your recommendations regarding the improvement of the discussion section and the exploration of future perspectives on how to apply these findings in diagnostic and practical fields. We fully agree and will make the following improvements accordingly. We will add a dedicated paragraph in the discussion section to explore the potential application of these findings in diagnostics and clinical practice. Specifically, we will address the following points. Potential as Diagnostic Biomarkers: Based on our findings regarding the activation of factors related to the MAPK signaling pathway and the expression changes of MMPs and TIMPs, we will discuss their potential as diagnostic biomarkers. Development of Therapeutic Targets: We will elaborate on the potential use of MAPK pathway inhibitors in modulating inflammatory responses and tissue repair. Future Research Directions: We will propose future studies that can further investigate the role of these pathways in early disease diagnosis and intervention(Page 14, Line 404-418). We will carefully review the entire manuscript to ensure that all necessary elements (e.g., E. coli) are correctly italicized, in accordance with academic writing standards. Thank you for your insightful suggestions. We will use this opportunity to further enhance the quality and applicability of our manuscript. If you have any additional comments or suggestions upon reviewing the revised manuscript, we are always open to them and will handle them with care.

Round 2

Reviewer 1 Report

Comments and Suggestions for Authors

No other comments

Author Response

Dear Reviewer, Thank you very much for your positive feedback and the time you have spent on our manuscript. We are pleased to hear that you have no other comments, and we hope that our work meets the standards of the journal. Best regards, Yuanyuan Zhang

Reviewer 2 Report

Comments and Suggestions for Authors

Second Round Review:

Peer review report on: “Impact of Escherichia coli and Lipopolysaccharide on the MAPK Signaling Pathway, MMPs, TIMPs, and the uPA System in Bovine Mammary Epithelial Cells”

Thank you for the corrected version, but some of the previously mentioned shortcomings are still present in the manuscript.

In the authors’ response: “we have revised the Materials and Methods section to include precise details, such as the number of experimental animals, technical replications, primer sequences with melting temperatures, apparatus identification, inhibitor usage, and a clear description of treatments (e.g., LPS + E. coli).”

Nevertheless, in the revised version, the number of experimental animals (line 116) or the number of technical replications (qPCR, Western blot), the melting temperature of the primers (Table 1) are still not mentioned in the Materials and methods section.

Other concerns:

Line 138: Was DNase treatment performed during RNA isolation?

Line 139: How did you checked the RNA integrity after isolation? How did you measure the RNA concentration and purity?

Line 145: How many reference genes did you test? How did you choose the most stable reference genes?

Line 170: The reference of the comparative 2-ΔCT method is missing.

Line 168: Which software did you used to densitometric analysis?

In the results of Western blot, in addition to the specific band of some target proteins, non-specific bands can be seen on the blots. Could be due to incomplete blocking or non-specific binding of antibodies? This phenomenon can reduce the accuracy of the results.

Third Round Review:

Thank you for your responses.

I have only two comments on the answers:

Response 1: “ The melting temperature of the primers were mentioned during the PCR cycle. (Line 406-407)”   

Comment: In the PCR temperature cycle, the annealing temperature should be specified, not the melting temperature of each primer.

The melting temperature of the primers should be indicated in the Table 1, separately for each primer, as it depends on the sequence of primers.

Here is an example: https://doi.org/10.3390/ijms25179187

Response 3: The concentration and purity of RNA were determined using a microplate reader. The microplate reader was used to measure the OD values (A260/A280) and concentration of RNA. When the RNA concentration is between 500 and 1000 ng/μl, and the OD value (A260/A280) is between 1.8 and 2.0, both the concentration and purity are considered to be within the acceptable range.(Line 401-405).

Comment: The part on measuring RNA integrity is missing from your answer. Did you check the RNA integrity with e.g. agarose gel electrophoresis or Qubit or Bioanalyzer?

Author Response

Second Round Response:

Comments 1: The number of experimental animals (line 116) or the number of technical replications (qPCR, Western blot), the melting temperature of the primers

(Table 1) are still not mentioned in the Materials and methods section.

Response 1: The number of experimental animals is unknown. The cells used in this experiment were isolated and cryopreserved in our laboratory in advance. No animals were involved in this study. This has been corrected in the manuscript. The experiment was performed with three technical replicates. The melting temperature of the primers were mentioned during the PCR cycle.(Line 406-407)

Comments 2: Line 138: Was DNase treatment performed during RNA isolation?

Response 2: DNase treatment was performed during the RNA extraction procedure. The kit used includes a step to remove DNA contamination.

Comments 3: Line 139: How did you checked the RNA integrity after isolation? How did you measure the RNA concentration and purity?

Response 3: The concentration and purity of RNA were determined using a microplate reader. The microplate reader was used to measure the OD values (A260/A280) and concentration of RNA. When the RNA concentration is between 500 and 1000 ng/μl, and the OD value (A260/A280) is between 1.8 and 2.0, both the concentration and purity are considered to be within the acceptable range.(Line 401-405)

Comments 4: Line 145: How many reference genes did you test? How did you choose the most stable reference genes?

Response 4: In our laboratory, which focuses on research related to bovine mastitis, the reference gene we selected is one that has been consistently used by our lab over time.

Comments 5: Line 170: The reference of the comparative 2-ΔCT method is missing.

Response 5: In our study, we applied the 2-ΔCT method to normalize the expression of target genes to an endogenous reference gene and to compare the expression levels between different samples. We have now added the appropriate reference to our manuscript to provide proper attribution to the original work that introduced this method.(Line 434)

Comments 6: Line 168: Which software did you used to densitometric analysis?

Response 6: The densitometric analysis was performed using ImageJ in this experiment.(Line 435)

Comments 7: In the results of Western blot, in addition to the specific band of some target proteins, nonspecific bands can be seen on the blots. Could be due to incomplete blocking or non-specific binding of antibodies? This phenomenon can reduce the accuracy of the results.

Response 7: In the analysis of Western blot bands, background interference and the influence of non-specific bands were eliminated, which will not affect the experimental results.

Thank you very much for your valuable suggestions. I have made the revisions in the manuscript accordingly. I wish you a pleasant and productive day at work.

Third Round Response:

Comment 1: In the PCR temperature cycle, the annealing temperature should be specified, not the melting temperature of each primer.

The melting temperature of the primers should be indicated in the Table 1, separately for each primer, as it depends on the sequence of primers.

Response 1:The melting temperature of each primer has been added in Table 1.

Comment 2: The part on measuring RNA integrity is missing from your answer. Did you check the RNA integrity with e.g. agarose gel electrophoresis or Qubit or Bioanalyzer?

Response 2:The integrity of RNA was not tested in this experiment. If you think this part is important, I can perform agarose gel electrophoresis to verify the integrity of RNA. The results of RNA integrity validation by agarose gel electrophoresis are as follows.

Thank you for your valuable comments. I have completed the revisions in the manuscript and would appreciate your confirmation. Wishing you a pleasant life and work.

Reviewer 3 Report

Comments and Suggestions for Authors

The authors upgraded the quality of the manuscript as requested

Comments on the Quality of English Language

the qualityt was upgraded

Author Response

(The authors gave the same response as above.)
